# AddictedChem: A Data-Driven Integrated Platform for New Psychoactive Substance Identification

**DOI:** 10.3390/molecules27123931

**Published:** 2022-06-19

**Authors:** Mengying Han, Sheng Liu, Dachuan Zhang, Rui Zhang, Dongliang Liu, Huadong Xing, Dandan Sun, Linlin Gong, Pengli Cai, Weizhong Tu, Junni Chen, Qian-Nan Hu

**Affiliations:** 1CAS Key Laboratory of Computational Biology, Shanghai Institute of Nutrition and Health, University of Chinese Academy of Sciences, Chinese Academy of Sciences, Shanghai 200031, China; hanmengying2018@sibs.ac.cn (M.H.); liusheng2019@sibs.ac.cn (S.L.); zhangdachuan2017@sibs.ac.cn (D.Z.); zhangrui2019@sibs.ac.cn (R.Z.); liudongliang2018@sibs.ac.cn (D.L.); xinghuadong2019@sibs.ac.cn (H.X.); sundandan2017@sibs.ac.cn (D.S.); gonglinlin2018@sibs.ac.cn (L.G.); caipengli@picb.ac.cn (P.C.); 2Tianjin Institute of Industrial Biotechnology, Chinese Academy of Sciences, Tianjin 300308, China; 3Wuhan LifeSynther Science and Technology Co., Limited, Wuhan 430000, China; tuweizhong@163.com (W.T.); service@esynbio.com (J.C.)

**Keywords:** drug addiction, database, new psychoactive substance, machine learning, prediction

## Abstract

The mechanisms underlying drug addiction remain nebulous. Furthermore, new psychoactive substances (NPS) are being developed to circumvent legal control; hence, rapid NPS identification is urgently needed. Here, we present the construction of the comprehensive database of controlled substances, AddictedChem. This database integrates the following information on controlled substances from the US Drug Enforcement Administration: physical and chemical characteristics; classified literature by Medical Subject Headings terms and target binding data; absorption, distribution, metabolism, excretion, and toxicity; and related genes, pathways, and bioassays. We created 29 predictive models for NPS identification using five machine learning algorithms and seven molecular descriptors. The best performing models achieved a balanced accuracy (BA) of 0.940 with an area under the curve (AUC) of 0.986 for the test set and a BA of 0.919 and an AUC of 0.968 for the external validation set, which were subsequently used to identify potential NPS with a consensus strategy. Concurrently, a chemical space that included the properties of vectorised addictive compounds was constructed and integrated with AddictedChem, illustrating the principle of diversely existing NPS from a macro perspective. Based on these potential applications, AddictedChem could be considered a highly promising tool for NPS identification and evaluation.

## 1. Introduction

Drug addiction is a chronic relapsing disease that involves drug sourcing and continuous use and adversely affects the brain. Individuals with drug addiction represent all age groups. Drug addiction often causes serious physical and mental damage to the individual, leading to societal problems, such as rising social crime rates and economic losses. The 2021 World Drug Report of the United Nations Office on Drugs and Crime (UNODC) indicated that approximately 275 million people took drugs worldwide in 2020, whereas more than 36 million individuals had drug use disorders [1]. Consequently, specific regulations have been introduced to control addictive compounds. The United Nations formulated three United Nations conventions in 1961 [2], 1971 [3], and 1988 [4] to classify addicted drugs and precursors. The European Union has no legislation to classify narcotic or psychotropic substances, while it has a pan-European control for rapid detection. For the USA, the Controlled Substances Act is a statute of the US Drug Enforcement Administration (DEA), which establishes federal US drug policy under which the manufacture, importation, possession, use, and distribution of certain substances is regulated [5]. Depending on the risk of dependence and abuse, and addiction likelihood, controlled substances are divided into Schedule I–V levels, where level I comprises research drugs whose use for medical purposes is strictly prohibited, and drugs in the other four categories can be used for medical treatment.

Most controlled compounds are agonists of receptors of the human central nervous system, such as opioid receptors, cannabis receptors, 5-hydroxytryptamine receptors, acetylcholine receptors, dopamine receptors, and γ-aminobutyric acid receptors [6]. The specific addiction mechanism may be related to the reward circuit mechanism in the brain [7]. For example, morphine stimulates μ receptors located on γ-aminobutyric acid neurons in the ventral tegmental area (VTA), thus disinhibiting dopaminergic neurons. This effect consequently promotes dopamine release from the VTA to the nucleus accumbens, triggering a rewarding effect [8,9]. However, the specific mechanism of addiction to most controlled substances remains unclear [10], and different levels of information must be integrated for in-depth research into controlled substances. However, a large amount of experimental data on controlled compounds exists in the scientific literature and databases. For example, the DrugBank database [11] contains information on the physiological targets and pharmacological effects of some controlled compounds; admetSAR [12] contains experimental and predicted values of the absorption, distribution, metabolism, excretion, and toxicity (ADMET) of controlled compounds; BindingDB [13] provides comprehensive information about protein–target interactions related to controlled compounds; and the Comparative Toxicogenomics Database (CTD) [14] provides detailed information on the interaction of controlled substances with genes and molecular pathways. To date, no single database provides experimental biologists with comprehensive information on controlled substances.

In addition to controlled substances *per se*, their analogues, i.e., new psychoactive substances (NPS), are constantly devised [15]. NPS are molecules that exert strong psychoactive effects mimicking the effects of legal recreational drugs and are difficult to detect by routine drug screening [16]. The UNODC first used the term ‘new psychoactive substances’ to refer to ‘substances of abuse, either in a pure form or as a preparation, that are not controlled by the 1961 Single Convention on Narcotic Drugs or the 1971 Convention on Psychotropic Substances, but which may pose a public health threat’ [17]. For example, a synthetic stimulant called ‘bath salt’ is a cathinone analogue [18]. At the end of 2020, approximately 830 NPS were monitored by the European Monitoring Centre for Drugs and Drug Addiction (EMCDDA), and their numbers continue to increase [15]. How to classify NPS from the many substances is an important issue that is challenging. Within drug samples, forensic and toxicological analyses can detect, identify, and quantify NPS. However, without a certified sample, NPS can be screened using methods such as high-resolution mass spectrometry [19]. The above efforts to identify and predict NPS are cost- and time-consuming. From a computational viewpoint, a quantitative structure–activity relationship model was used to verify the analogues of amphetamine cathinone [20]. We still need more computational tools to produce low-cost, rapid predictions of NPS and to pave the way for the earlier identification of emerging drugs.

In this study, we present AddictedChem, which is a scientific database of controlled substances. The database integrates multiple dimensions of data, allowing the analysis of controlled substances from the perspectives of molecules, targets, and functional enrichment, facilitating the discovery of new targets and drugs for the treatment and prevention of addictive diseases. We also created a consensus model to predict NPS to accelerate experimental NPS identification. Finally, we visually explored the chemical space of addictive compounds and discovered the spatial distribution patterns of NPS and controlled substances, providing the user with an interactive online exploration platform (http://design.rxnfinder.org/addictedchem/chemical_space). AddictedChem will facilitate further addictive compound research and identification.

## 2. Results and Discussion

### 2.1. Overview of AddictedChem

The developed analysis platform comprises two parts (Figure 1): the knowledge base of 622 controlled substances (classified as shown in Figure 2A) and the NPS prediction platform. Five search methods, namely, text query, structure, similarity, maximum common substructure (MCS), and fragment, are available in the knowledge base.

The user can directly enter the name or structure of a controlled substance to obtain the relevant literature, target, ADMET, gene, pathway, and bioassay information (Figure 2B). The text query method retrieves controlled substances using synonymous names from PubChem (see Materials and Methods for details).

The structure search method requires the user to provide the compound in the simplified molecular input line entry specification (SMILES) format or use the JSME tool [21] to draw the structure of the compound of interest. The JSME tool automatically converts the 2D molecular image into the SMILES format. The similarity retrieval method is based on a similarity algorithm [22], which returns the 20 closest molecular structures of the compound searched in the AddictedChem database. The MCS retrieval method is based on the fMCS algorithm [23] and returns 20 compounds with the largest substructure of the searched compound available in the AddictedChem database. Finally, the fragment retrieval method is based on the theory of connected subgraphs [24]. After entering the chemical structure of a compound, all fragment structures of the compound are retrieved.

In the NPS prediction platform, the user obtains the result report after entering the compound in the SMILES format or several compounds formatted in rows. The results are presented as three information sets: (1) consensus model results; (2) visualised results of 29 models based on seven descriptors, to be intuitively viewed by the user; and (3) specific prediction results of the 29 models in a list format. The website allows researchers to easily obtain prediction results.

The addictive compounds are still far from saturation because more new psychoactive substances are emerging and have not been regulated completely. Because AddictedChem is an ongoing curation, it is continuously maintained and updated as new controlled substances are published. We will update the database annually to improve the comprehensiveness of data and collect more information about emerging controlled substances from more countries and regions in the future.

### 2.2. Scaffold Analysis of Controlled Substances

The analysis of 622 controlled substances using Murcko scaffolds method yielded 245 scaffolds. Figure 2C shows the 10 most common scaffolds in controlled substances from different schedules. The top five scaffolds were benzene (11.09%), 1,3-dihydro-5-phenyl-1,4-benzodiazepin-2-one (3.54%), 4-phenylpiperidine (3.38%), 4-anilino-*N*-phenethyl-piperidine (3.38%), and acyclic (2.25%). The benzene ring is a common core scaffold in many chemical databases [25]. 1,3-Dihydro-5-phenyl-1,4-benzodiazepin-2-one is found in flurazepam, lorazepam, and oxazepam; 4-phenylpiperidine is mainly found in betameprodine, betaprodine, and alphaprodine; and 4-anilino-*N*-phenethyl-piperidine is mainly found in fentanyl, ocfentanil, and isobutyryl fentanyl, and it is the direct precursor of fentanyl and certain fentanyl analogues (such as acetylfentanyl).

We then used the Murcko scaffolds method to analyse controlled substances classed into different schedules (Appendix A). The analysis revealed that 1,3-dihydro-5-phenyl-1,4-benzodiazepin-2-one is mainly found in Schedule IV drugs, 4-phenylpiperidine is mainly found in Schedule I and II drugs, and 4-anilino-*N*-phenethyl-piperidine is mainly found in Schedule I drugs. Notably, the top three scaffolds of Schedule III drugs are the same isomer, indicating that the structures of the Schedule III drugs are the most similar. These findings showed the diversity of the scaffolds of controlled substances, but the scaffolds of controlled substances with the same schedule are similar.

### 2.3. Analysis of Controlled Substance Targets

We then analysed controlled substance targets and used the PubChem ID of a compound of interest as the basis to search for its target information in the BindingDB database [13]. Analysis revealed that as of 4 January 2021, only 144 controlled substances (of 622) were associated with 104 (after deduplication) target studies, which comprised 475 compound–target pairs (Figure 3).

Considering the binding drug targets (Figure 2D), the most common target of the identified 144 compounds was the Mu-type opioid receptor (UniProt ID: P35372), targeted by 26 compounds; followed by the delta-type opioid receptor (UniProt ID: P41143), targeted by 25 compounds; and the Kappa-type opioid receptor (UniProt ID: P41145), targeted by 24 compounds. While these three targets all interact with Schedule I–IV drugs, they mostly interact with Schedule II drugs. The fourth most common target was the sodium-dependent serotonin transporter (targeted by 19 compounds), which mainly interacts with Schedule I drugs. The other common targets were canalicular multi-specific organic anion transporter 2 (targeted by 18 compounds) and multidrug resistance-associated protein 4 (also targeted by 18 compounds), which mainly interact with Schedule IV drugs. Based on drug classification and statistical analysis of controlled substances from different schedules (Appendix A), Schedule II drugs had the highest degree of polymerisation, which was mainly because most of these compounds target cannabinoid receptors.

Considering the compound type (Figure 2E), the controlled substances with the greatest number of targets were 3,4-methylenedioxymethamphetamine (MDMA), 25I-NBOMe, chlordiazepoxide, acetaminophen (APAP), and lacosamide. MDMA, as a Schedule I drug, induces the reversal of human serotonin transporters, leading to serotonin release [26]. Another Schedule I drug, 25I-NBOMe, is a synthetic hallucinogen that is a derivative of the 2C-I family of substituted phenethylamines. The main receptor of 25I-NBOMe is the human 5-HT_2A_ receptor, and the compound interacts weakly with other receptors. Chlordiazepoxide, a Schedule IV drug, is a sedative and hypnotic drug from the benzodiazepine class. It interacts with GABA_A_ receptors/ion channels, influencing the central nervous system [27]. APAP, a Schedule IV drug, is a common analgesic and antipyretic drug that acts on many targets. Finally, lacosamide, a Schedule V drug, is an anticonvulsant compound. Controlled substances always activate or inhibit key targets in the signal transduction of the central nervous system; hence, identifying these targets helps to discover the mechanism of addiction.

### 2.4. Functional Enrichment Analysis

The analysis results revealed that pathway information was available for 166 controlled substances with 985 compound–pathway combinations. For the functional pathways of controlled substances, we observed that pathway information was available for 122 controlled substances with 209 compound–pathway combinations. Of the small number of the controlled substances (122), nearly 37% are related to the GABA-A receptor agonists/antagonists, which is followed by opioid receptor agonists/antagonists (Figure 2F). This is in line with the mechanism by which most regulated substances target receptor agonists or inhibitors.

However, it is worth noting that because of data deficiency, the pathway analysis was conducted only on controlled substances that have reported targets and pathways and did not take dose effects into consideration.

Likewise, we identified 261 compounds and 442 related genes in humans. We counted 5637 gene ontology (GO) annotations for these genes, of which 203 were significantly enriched, and we identified the top 40 GO annotations that were significantly enriched (Appendix A). GO analysis revealed that genes that interact with controlled substances were significantly enriched in the steroid metabolic process, response to the drug, response to oxidative process, and the metabolic process of fatty acids. Furthermore, biological process (BP) accounted for 90% of the 203 enriched GO annotations, indicating that addictive compounds are mainly involved in biological processes. These results reveal that controlled substances are widely involved in metabolic pathways in the human body and play a role in biological processes.

### 2.5. Model Performance for NPS Prediction

Using seven types of molecular descriptors and five machine learning algorithms, we built 29 models to predict potential NPS. The model’s performance on external validation and test sets is summarised in Table 1.

As shown, most models performed well with the test and external validation sets. Model performance with the test set was better than that with the external validation set. Because the test set was extracted from and constituted 20% of the entire dataset, the external validation set was collated using external data for verification. Among the seven molecular feature description methods, Morgan fingerprint performed the best with the four traditional machine learning models with the test set. The performance of SECFP with the NB, RF, and SVM models with the validation set was better than with the test set.

Meanwhile, the performance of the deep learning model was not better than traditional machine learning. The deep learning model performed well with the test set, but its performance with the external validation set was average. Among traditional machine learning models, the RF model performed the best, which was followed by SVM, LR, and NB. The best performance with the test set was obtained with the RF model based on MACCS (BA of 0.940 and AUC of 0.986) (Table 1). The best performance with the external validation set was obtained with the NF model based on SECFP (BA of 0.919 and AUC of 0.968). However, as shown in Table 1, some models including NB::Mol2vec, LR::RDkit, RF::MHFP6, and ChemBERT have extremely poor performance. We deduced that model overfitting might have caused some models to perform much worse on the external validation set than on the test set. The test set and the training set belong to the same source, although the compounds are different. Compared with the test set, the external validation set is a completely different source. Furthermore, the number of externally labelled validation sets is very small. These reasons may cause some overfitted models to perform poorly on external validation sets. Hence, we selected RF::MACCS and NB::SECFP from the 29 models to form a consensus model. For intuitive comparison, we further compared the receiver-operating characteristic (ROC) and PR curves of all these methods; in the machine learning algorithm, we used Morgan fingerprint because it was very stable in all models (Figure 4). The analysis revealed the necessity of constructing a consensus model that retained the best model performance and feature description to the greatest extent.

The consensus scoring model is used for the online classification prediction function of addictive compounds. Deoxymethoxetamine (DXME) is a hallucinogen from the arylcyclohexylamine family that exerts dissociative effects. It has emerged on the illicit market since October 2020 and was first identified by a Danish forensic laboratory in February 2021 [28]. As a newly included NPS by HighResNPS (https://highresnps.forensic.ku.dk, accessed on 27 April 2022), DXME does not appear in our datasets. We predicted DXME using the online prediction function freely available at http://design.rxnfinder.org/addictedchem/prediction/. Our consensus model correctly predicted it as an addictive compound (Appendix A). This finding, to some extent, indicates that our model can predict potential NPS.

Despite our best efforts to collect compound addiction datasets, the datasets collected in this study are still small. This may affect the model’s ability to discriminate structurally unique novel addictive compounds. In the future, we will try new machine learning algorithms to model methods, including positive unlabelled learning and few shot learning, to further expand the generalisation ability and robustness of the model.

### 2.6. Exploration of the Chemical Space of Addictive Compounds

To show the chemical space of addictive compounds more intuitively, we created vectorisations of all the compounds used in the mathematical method and explored their spatial distribution patterns. The TMAP (tree-map) [29] visualisation result (Figure 5) shows the two-dimensional distribution of molecules on the tree and reflects similarities calculated in the high-dimensional MHFP6 space. The compounds in schedules of controlled substances (red) and abused drugs listed by the Cayman Chemical Company (blue) were positive samples, and DrugBank (green) and Generally Recognised as Safe (GRAS) (purple) compounds were negative samples (Figure 5). The visualisation shows that the positive and negative classes clustered (Figure 5), and abused drugs from the Cayman Company list surrounded the compounds in schedules of controlled substances. On a local magnification scale, some addictive analogues surrounded the same controlled compound, such as the controlled substance methanone (inset, Figure 5). Its analogues undergo some simple changes, such as a single terminal bond becoming a double bond, a bond change, or the addition of F, Cl, or Br atoms. As shown in Figure 5, most areas of chemical space around controlled substances (red) remain unexplored, which means that there are still many possibilities for NPS to be designed. Thus, expanding the chemical space may accelerate potential NPS discovery research. The user can visit the interactive version to interactively experience the space of addictive compounds.

## 3. Materials and Methods

### 3.1. Data Sources

A detailed summary of all datasets used in this study is shown in Appendix A. To construct a machine learning model for the classification of addictive compounds, we collected compound data from various publicly available sources. We carefully screened 854 data sources from PubChem [30], which is the world’s largest collection of freely accessible chemical information to acquire many addiction-related compounds. Among them, substances controlled under the laws and regulations of the United States are listed by the DEA. It provides a relatively comprehensive list and is easily accessible. Furthermore, the Cayman Chemical Company is known as a provider of reference standards to crime labs. Therefore, the positive samples were finally derived from the list of controlled substances by the DEA and a list of abused drugs from Cayman Chemical Company. The compounds in the first list were divided into five schedules according to risk, abuse, and addiction. Schedules I–V contain 622 compounds, with 259 in Schedule I; 102 in Schedule II; 138 in Schedule III; 114 in Schedule IV; and 9 in Schedule V. The second list contains 1880 compounds. Because many of its components are NPS, these compounds were included as positive samples. Before the analysis, controlled substances were removed from that list. The negative samples, which contain compounds considered non-addictive, were also derived from two lists. The first is a list of GRAS compounds from the US Foods and Drug Administration. The compound names were converted to a SMILES [31] format by using PubChemPy (https://pubchempy.readthedocs.io/en/latest/, accessed on 31 December 2021), and the compounds without SMILES were deleted, yielding 173 compounds. The second list was from DrugBank [11]. To avoid the problem of sample imbalance, the ratio of positive to negative samples was set to 1:1. Therefore, 2329 compounds were finally randomly selected from the second list after removing controlled substances, NPS, and GRAS compounds. The above datasets were used for model training and testing.

The external validation sets for model evaluation are derived from two main sources (Appendix A). The 2017–2020 NFLIS-Drug substance list [32] provided by DEA constituted positive sample data for the external validation set and contained 184 substances. The NFLIS-Drug contains the results of drug chemistry analyses and associated information from drug cases. Negative samples for the external validation set were 149 newly approved drugs retrieved from the Kyoto Encyclopaedia of Genes and Genomes (KEGG) [33] for the period 2017–2021.

In addition to modelling-related data, there are more datasets for data analysis and database construction (Appendix A). Basic physical and chemical attribute information and bioassay information for controlled substances were obtained from PubChem [30], which was searched using a list of controlled substances obtained from the DEA. The literature on controlled substances was obtained from the literature section in PubChem, which was based on the individual compound interface. ADMET data were obtained from admetSAR [12], which were divided into experimental and predicted data categories on chemicals. The data on controlled substance targets were obtained from BindingDB [13]. Interactions between the controlled substances and pathways and GO annotations originate from CTD [14]. The metabolic pathway data of controlled substances were acquired from the small-molecule pathway database (SMPDB) [34], which is a great source of metabolic information for many psychoactive drugs.

### 3.2. Data Curation

For subsequent modelling and database construction, we processed all compounds obtained from sources to maintain the quality of the final data. We used the open source cheminformatics toolkit RDKit [35] (http://www.rdkit.org) to evaluate the SMILES of compounds for validity and to discard invalid molecules. We then normalised the effective compound structures using RDKit.

### 3.3. Data Analysis

#### 3.3.1. Scaffold Analysis of Controlled Substances

Murcko scaffolds [36] were used to analyse the molecular characteristics of the controlled substances and explore their diversity. The Murcko scaffolds calculation method involves retaining the ring structure of the compound and the linker between the ring structures and removing all the side chains on the ring structure. The Murcko scaffolds algorithm was implemented based on RDKit [35], which is a chemical information toolkit.

#### 3.3.2. Target Analysis of Controlled Substances

Cytoscape 3.7.0 [37] was used to comprehensively evaluate the targets of controlled substances to analyse the relationship between controlled substances classified at different addiction levels and their targets. The controlled substance list was from the DEA. The control substance targets were from BindingDB [13]. The relationship between controlled substances and their effects from the perspective of compounds and targets was statistically analysed using Python.

#### 3.3.3. Functional Enrichment Analysis

To evaluate the functional genes targeted by controlled substances, pathway enrichment and GO annotation analyses were performed. The MeSH ID [38] of the controlled substances was a standard to retrieve pathway information related to controlled substances from CTD [14]. That is because the database already contains the information annotated from the literature on the relationship between manually annotated compounds and genes or proteins. There are two main parts in pathway analysis: one part is that, according to the manual annotated compound and gene data list in the CTD database, the R package clusterProfiler [39] was employed to conduct KEGG pathway enrichment analysis using compound-associated genes, and we considered only the pathways with *p* values lower than 0.1. The other part is that because the first part can be enriched only from the first to sixth categories (1. Metabolism–6. Human Diseases) and does not contain key functional pathways (7. Drug Development), we used Python to retrieve the seventh category in the KEGG pathway for controlled substances. Because the seventh pathway designs the mechanism of addiction, we performed a functional summary of the seventh pathway data.

The clusterProfiler package [39] was used for GO annotation analysis of gene sets related to controlled substances.

### 3.4. Model Construction

Seven molecular feature representation methods were used to characterise compounds of interest, including RDKit descriptors and E-state [40], MACCS [41], RDKit fingerprint, Morgan fingerprint [42], SMILES-extended contiguity fingerprint (SECFP) [43], MinHash fingerprint (MHFP6) [43], and Mol2vec [44] (data dimensions: 279, 166, 2048, 2048, 2048, 2048, and 300, respectively). The first six of these methods use RDKit, whereas Mol2vec uses an unsupervised machine learning method based on Python.

Four machine learning methods, logistic regression (LR) [45], naive Bayes (NB) [46], random forest (RF) [47], and support vector machine (SVM) [48] were adopted, which were implemented based on scikit-learn [49]. The training set and test set were randomly generated (ratio: 8:2). In addition, a deep learning method model, ChemBERTa_zinc250k_v2_40k from ChemBERTa [50], was used for pre-training. It was then fine-tuned for 10 epochs using the generated dataset. The dataset was divided into a training set, a test set, and an internal validation set in a ratio of 8:1:1.

After creating 29 models based on seven molecular feature representation methods and five machine learning algorithms, two models that performed the best with the external validation set and the test set were selected to create a consensus model for NPS prediction. To ascertain highly rigorous NPS prediction, the integrated model predicted a positive result when either of the two models yielded a positive result and a negative result when both models predicted a negative result.

### 3.5. Performance Evaluation

In this study, test and external validation sets were used to evaluate the model performance based on the following parameters: true positive (*TP*), false negative (*FN*), true negative (*TN*), false positive (*FP*), sensitivity (*SE*), specificity (*SP*), balance accuracy (*BA*), and Matthew’s correlation coefficient (*MCC*), as defined below (Equations (1)–(7)).
(1)$$\begin{array}{c} SE=\frac{TP}{TP\;+\;FN} \end{array}$$
(2)$$\begin{array}{c} SP=\frac{TN}{TN\;+\;FP} \end{array}$$
(3)$$\begin{array}{c} BA=\frac{1}{2}\times \left(SE+SP\right) \end{array}$$
(4)$$\begin{array}{c} MCC=\frac{TP\;\times \;TN\;-\;FP\;\times \;FN}{\sqrt{\left(TP\;+\;FP\right)\left(TP\;+\;FN\right)\left(TN\;+\;FP\right)\left(TN\;+\;FN\right)}} \end{array}$$
(5)$$\begin{array}{c} Precision=\frac{TP}{TP\;+\;FP} \end{array}$$
(6)$$\begin{array}{c} Recall=\frac{TP}{TP\;+\;FN} \end{array}$$
(7)$$\begin{array}{c} F1=\frac{2\;\times \;Precision\;\times \;Recall}{Precision\;+\;Recall} \end{array}$$

The *F*1-measure (*F*1) score, ROC curve, and area under ROC (AUC) were also used to evaluate model performance.

### 3.6. Chemical Space Exploration

To explore the chemical space of addictive compounds, we visualized the chemical space occupied by all molecules in the model building datasets using TMAP (tree-map) [29], which is capable of representing large datasets and arbitrary high dimensionality as a two-dimensional tree. We considered the SMILES of all compounds as input and computed their MHFP6 molecular fingerprints [43]. Then, LSH Forest [51] was used to generate indexes for fast k-nearest neighbour [52] retrieval. Furthermore, the data points indexed in the LSH forest were used to generate an undirected weighted graph of c-approximate k-nearest neighbour (c-k-NNG). We applied Kruskal’s algorithm to construct a minimum spanning tree on the weighted c-k-NNG. Finally, the Python package Faerun [29] was used to visualise the data and show the resulting MST. The interactive version results are freely available at http://design.rxnfinder.org/addictedchem/chemical_space.

### 3.7. Database and Webserver

The comprehensive addicted compounds portal comprises a front-end user interface and a back-end database. The data used in the construction of the database mainly include those of controlled substances from DEA, basic physical and chemical attributes, bioassays for controlled substances from PubChem and literature on controlled substances from PubChem [30] and PubMed, experimental and predicted ADMET data from admetSAR [12], enriched associations between controlled substances and pathways from CTD [14], enriched associations between controlled substances and GO annotations from CTD [14], and metabolic pathway data of controlled substances from the SMPDB [34]. All of the above information is presented to the user through the search function module, which has five kinds of search methods. AddictedChem provides a browse function for all listed addictive compounds in our database. The user can either download the list of chemicals or follow the link to the compound details page. For the prediction of potential addictive compounds, we implemented an online prediction tool that uses a consensus scoring model formed with the RF::MACCS and NB::SECFP models. The entire project was implemented using Ubuntu (18.04.2), and the back-end framework used Django (2.2) based on Python (3.7.8). The data for the entire project are stored in MySQL (8.0.16). The ECharts (4.2.0) software was used as a graphical visualisation framework. The Bootstrap table (1.15.5) was used for static and dynamic data table display and relied on Bootstrap (3.3.7) and jQuery (2.1.1). Using the Chrome web browser is recommended to access and browse the database. AddictedChem is freely available to the research community at http://design.rxnfinder.org/addictedchem/. The users do not have to register or login to access the features of the databases. User-friendly online tutorials are provided to facilitate user hands-on operations at http://design.rxnfinder.org/addictedchem/help/. Finally, AddictedChem is an online platform outside the box, and the user can download all the data from http://design.rxnfinder.org/addictedchem/downloads/.

## 4. Conclusions

We constructed AddictedChem, a comprehensive database of controlled substances; analysed the existing data on molecular scaffolds, targets, gene-related pathways, and GO annotations; and developed a consistence predictive model based on multiple molecular description approaches and machine learning algorithms to assist NPS identification and analysis. Concurrently, we built a chemical space of addictive compounds, which revealed the chemical diversity of NPS. As the first step in the data-driven identification of potentially addictive molecules, this research can be applied to a wide range of scenarios, such as providing a preliminary assessment of drug addiction in the early stages of new drug development, identifying new addictive illegal drugs circulating on the market, and discovering potentially addictive components in food to improve food safety. In the next step, we will combine the prediction model with the metabolite identification model and untargeted metabolomic data to establish a more comprehensive analysis to better support the identification and detection of new illegal drugs. With these potential applications, we expect that AddictedChem will become an indispensable tool for use in future NPS studies.

## Figures and Tables

**Figure 1 molecules-27-03931-f001:**
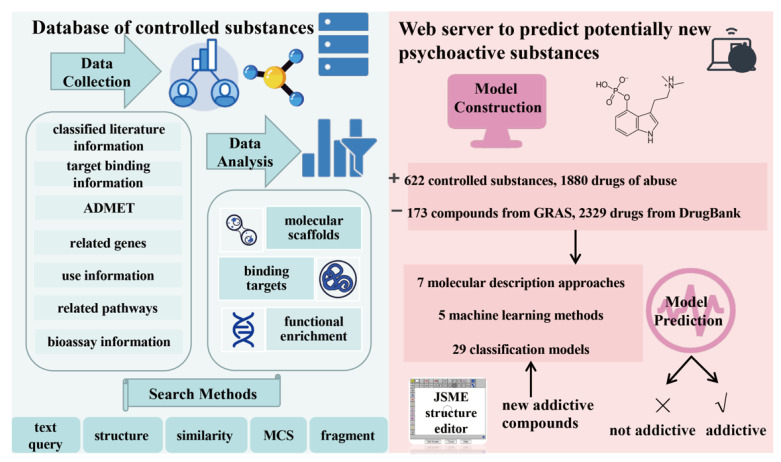
Flow chart of AddictedChem. (**Left**), multilevel database of controlled substances. (**Right**), NPS prediction platform.

**Figure 2 molecules-27-03931-f002:**
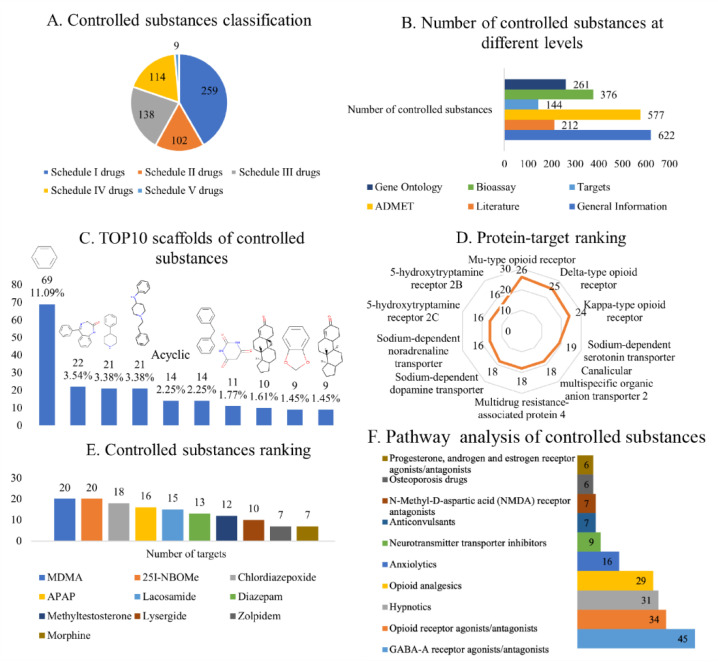
Statistical analysis of entries in AddictedChem. (**A**) Classification of controlled substances. (**B**) Number of controlled substances on different schedules. (**C**) Top 10 scaffolds of controlled substances, namely, benzene (11.09%), 1,3-dihydro-5-phenyl-1,4-benzodiazepin-2-one (3.54%), 4-phenylpiperidine (3.38%), 4-anilino-*N*-phenethyl-piperidine (3.38%), and acyclic (2.25%), diphenylmethane (2.25%), barbituric acid (1.71%), unnamed compound (1.61%), 1,3-benzodioxole (1.45%), unnamed compound (1.45%). The numbers indicate the frequency of occurrence of the scaffolds, and the percentage values are the ratio of every scaffold to 622 scaffolds. (**D**) Binding protein target ranking according to the number of controlled substances in the BindingDB database. (**E**) Controlled substances ranking according to the number of binding protein targets in the BindingDB database. (**F**) Pathway analysis of controlled substances. These pathways are from KEGG drug development class, and the numbers indicate the number of controlled substances.

**Figure 3 molecules-27-03931-f003:**
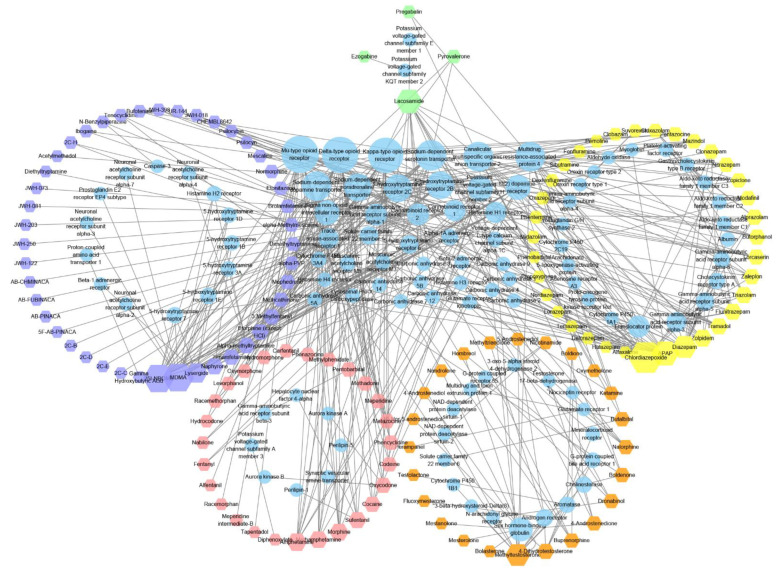
Map of controlled substances and their interaction targets. Hexagons represent controlled substances and circles are controlled substances. The hexagon and ellipse sizes represent the degree of connectivity. The circles formed by the shapes represent different levels of controlled substances and their targets. The circles arranged on a grid in the middle are targets that interact with multiple types of compounds. The number of compound targets gradually decreases, counter clockwise from the lowest point on the circle, as shown in the figure. Purple hexagons, Schedule I drugs; red hexagons, Schedule II drugs; orange hexagons, Schedule III drugs; yellow hexagons, Schedule IV drugs; green hexagons, Schedule V drugs. The figure was generated using Cytoscape (3.7.0).

**Figure 4 molecules-27-03931-f004:**
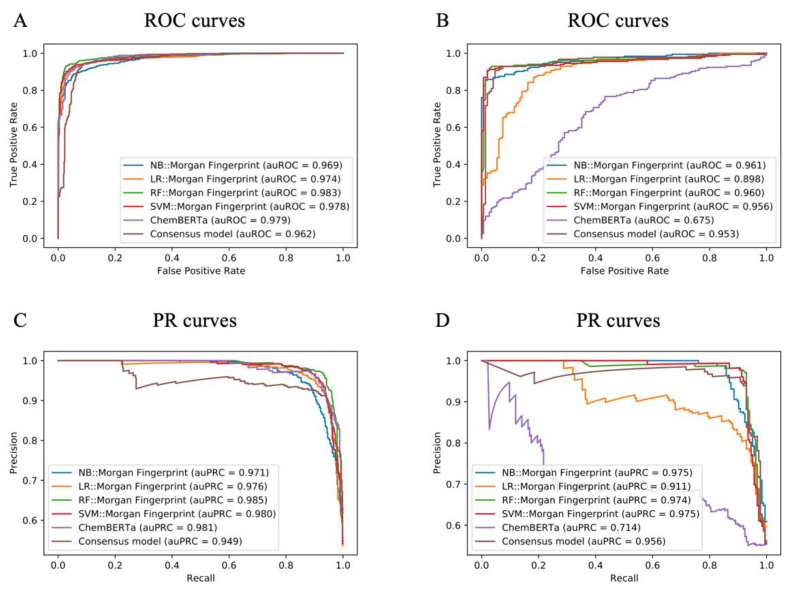
ROC curves and PR curves for models generated in the current study. (**A**) ROC curves for the test set. (**B**) ROC curves for the external validation set. (**C**) PR curves for the test set. (**D**) PR curves for the external validation set.

**Figure 5 molecules-27-03931-f005:**
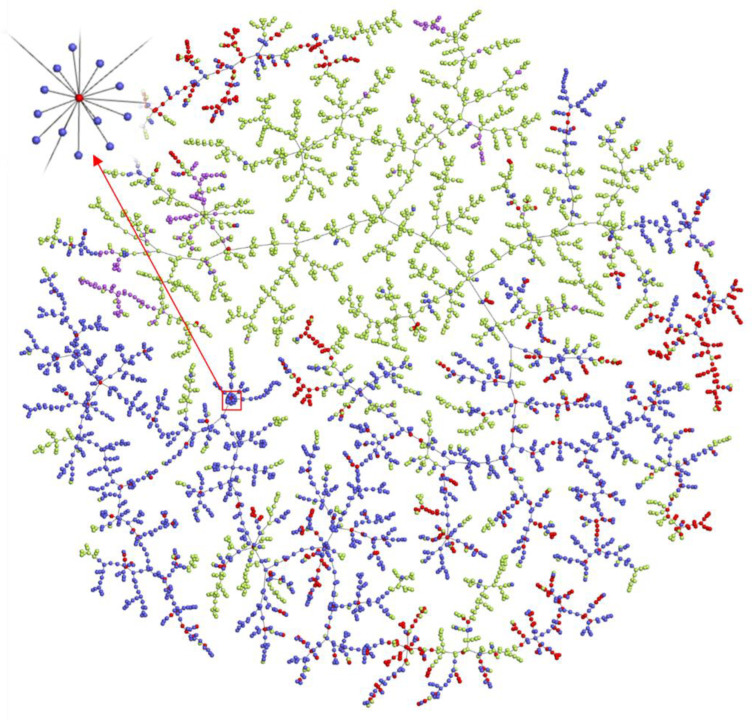
TMAP visualisation of model building datasets in the MHFP6 chemical space. The MHFP6 molecular fingerprints were used to characterise the compounds, generate indexes for fast k-nearest neighbour retrieval using LSH Forest, and visualise data using Faerun. Each data point in the figure indicates a chemical compound. The links connecting the data points represent the edges of the minimum spanning tree. All chemical compounds are shown in colour according to the dataset source. GRAS compounds (purple), DEA controlled substances (red), Cayman Chemical Company abused drugs (blue), and DrugBank compounds (green). In the upper left corner is seen a partial enlargement of methanone and the surrounding compounds. Please use the interactive version at http://design.rxnfinder.org/addictedchem/chemical_space to visualise the molecular structures associated with each point.

**Table 1 molecules-27-03931-t001:** Model performance with an external validation set and a test set.

		Test Set				External Validation Set			
		BA	MCC	F1	AUC	BA	MCC	F1	AUC
**NB**	RDKit Descriptors+E-state	0.810	0.627	0.817	0.897	0.780	0.572	0.820	0.858
	MACCS	0.794	0.595	0.803	0.887	0.806	0.629	0.844	0.895
	RDKit fingerprint	0.794	0.588	0.786	0.865	0.844	0.687	0.835	0.920
	Morgan fingerprint	0.909	0.818	0.905	0.969	0.911	0.817	0.912	0.961
	SECFP	0.886	0.772	0.882	0.960	0.919	0.834	0.921	0.968
	MHFP6	0.787	0.586	0.800	0.893	0.844	0.685	0.856	0.894
	Mol2vec	0.714	0.472	0.754	0.779	0.5	0.0	0.0	0.606
**LR**	RDKit Descriptors+E-state	0.899	0.797	0.897	0.966	0.656	0.310	0.670	0.727
	MACCS	0.894	0.787	0.891	0.962	0.741	0.480	0.760	0.837
	RDKit fingerprint	0.919	0.838	0.917	0.967	0.903	0.805	0.913	0.964
	Morgan fingerprint	0.931	0.862	0.929	0.974	0.837	0.683	0.864	0.898
	SECFP	0.921	0.842	0.919	0.969	0.888	0.775	0.899	0.931
	MHFP6	0.896	0.793	0.895	0.948	0.919	0.834	0.921	0.964
	Mol2vec	0.894	0.787	0.892	0.959	0.774	0.544	0.784	0.848
**RF**	RDKit Descriptors+E-state	0.944	0.888	0.942	0.985	0.815	0.640	0.847	0.887
	MACCS	0.940	0.880	0.938	0.986	0.892	0.786	0.906	0.950
	RDKit fingerprint	0.942	0.887	0.940	0.973	0.951	0.898	0.953	0.965
	Morgan fingerprint	0.950	0.900	0.948	0.983	0.921	0.842	0.929	0.960
	SECFP	0.942	0.884	0.940	0.979	0.943	0.885	0.948	0.964
	MHFP6	0.927	0.854	0.924	0.974	0.491	−0.019	0.574	0.500
	Mol2vec	0.921	0.842	0.918	0.973	0.702	0.478	0.795	0.871
**SVM**	RDKit Descriptors+E-state	0.919	0.838	0.916	0.975	0.806	0.629	0.844	0.858
	MACCS	0.921	0.842	0.918	0.980	0.860	0.720	0.875	0.927
	RDKit fingerprint	0.942	0.884	0.939	0.975	0.930	0.856	0.933	0.959
	Morgan fingerprint	0.933	0.870	0.929	0.978	0.928	0.854	0.934	0.956
	SECFP	0.934	0.871	0.931	0.977	0.939	0.874	0.942	0.960
	MHFP6	0.890	0.780	0.889	0.955	0.747	0.546	0.664	0.949
	Mol2vec	0.907	0.814	0.906	0.971	0.799	0.629	0.847	0.910
**ChemBERTa**	-	0.934	0.868	0.935	0.979	0.629	0.256	0.644	0.675

MHFP6, MinHash fingerprint; NB, naive Bayes; RF, random forest; SECFP, SMILES-extended contiguity fingerprint; SVM, support vector machine.

## Data Availability

AddictedChem is freely accessible at http://design.rxnfinder.org/addictedchem/. Classification models and relevant codes are available at https://doi.org/10.5281/zenodo.6508755. All the datasets in the project can be found at http://design.rxnfinder.org/addictedchem/downloads.

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
