# Peer review of "AddictedChem: A Data-Driven Integrated Platform for New Psychoactive Substance Identification"

_molecules, 2022, doi:10.3390/molecules27123931_

Round 1

Reviewer 1 Report

The idea of a database covering in depth information on both scheduled drugs and NPS is very interesting. The information it could provide would be very helpful for several stakeholders, like scientists, monitoring agencies, but also healthcare professionals.

The objective of this study should be clarified in the abstract and introduction, there seem to be three. One on controlled drugs (controlled where?), one on NPS (what exactly) and one on chemical space (what is this?). Also, it is unclear to me how this database was constructed (the methods needs clarification), and also, how it will remain up to date. Since data on a DEA list of drugs is (manually?) collected from several other databases up to 2020 and merged in this new database, this seems to be an issue, especially with all the NPS emerging each year and maybe also being controlled. Will they move from one side of the database (predict NPS) to the other (controlled substance)?

Finally, only merged / grouped data is presented in the article, but I assume the database is developed to query it for individual drugs. How would that look? What data would a user of the database receive?

Furthermore, the information in the introduction should be thoroughly checked, since incorrect information is included. See detailed comments on this below.

Abstract

Is it the first database? Please rewrite a bit.

Please include info on which substances (controlled substances where) the database is based, since control of substances varies strongly per country/continent.

Classified literature? How can it be included when classified?

Please clarify ‘to predict NPS online’: to predict what? The occurrence online by discussion of NPS in user fora, the occurrence of NPS in web shops, the occurrence of NPS via notification to monitoring bodies?

‘The potential chemical space’, what is this?

‘knowledgebase’, what is this?

Please be more specific in the abstract.

The database is meant to cover both typical illicit drugs, but also to identify NPS. This could be clearer in the title.

introduction

Line 37-40 please indicate if the numbers provided are worldwide numbers.

Line 45-48, this is not the case in other countries. Please rewrite and increase interest for non-US readers.

Line 53-54. Direct action on dopamine receptors is not the main mechanism by which morphine is addictive (which is stimulation of µ opioid receptors at GABA-neurons in the VTA). The listed reference also does not support this statement, please remove and adjust with correct references.

Line 69-70 please provide most recent data and do not present old data from 2018. The EMCDDA has reported already >800 NPS.

Line 71. While NPS have pharmacological activity, it often is not the same as the corresponding illicit drug. Please remove your statement ‘While the chemical structure of NPS is slightly different from that of controlled 71 substances, the pharmacological effect is preserved’, or support with references.

Line 73: There is no evidence that NPS in general are more addictive than typical recreational drugs. Please remove this statement or support with reference.

Line 79. This is not correct. The early warning system of the EMCDDA collects data from seized drugs (customs), forensics (fatal cases), poisonings, emergency department visits, etc. Please remove your text.

methods

Data sources. I do not understand how the database was constructed. Drugs were selected based on DEA lists and lists from Caymen Company (why?, how comprehensive is this?) Please clarify what databases were used for which endpoint (e.g., addiction (line 287?) or pharmacology targets etc.)

results clearly

figure 1: addicted should be addictive (Bottom right)

Figure 2D: protein target ranking. Please clarify. Does this only involve binding? Or are these functional targets?

Figure 2E: what targets? What is this figure displaying?

Figure 2F: Please define pathways. Is reversal of a monoamine reuptake transporter, or inhibition, a pathway in this figure? Since this appears to be the only functional (non binding) data in the database, I assume you have included transporters as an important pathway, but they are not listed.

Please separate kinetic pathways (e.g. metabolism) from dynamic pathways (effects). What I miss here is the main mechanism of action of many recreational drug. The inhibition (sometime also reversal) of monoamine reuptake transporters.

Apoptosis is found to be the most common used pathway, but there is no referral to dose. Apoptosis following recreational drugs exposure is often only seen at very high (millimolar) concentration, which are not relevant for human exposures. When identifying functional pathways, it is key to include dose in the analysis, otherwise faulty conclusions can be drawn (for example, in the current version is written: nearly 40% are related to the apoptosis pathway, which is consistent with the notion that addictive compounds may cause neuronal apoptosis. The is dose-dependent and at relevant doses, most drugs do not result in apoptosis)

Is the number of pathways (described in legend) or the number of compounds (line 202) listed?

Figure 3: the idea is interesting. But the way this is presented is not very useful. It is very difficult to read due to all the lines and difficult to interpret. Is there another way to present it? Also would there be a way to identify the main mechanism of action of drugs and include that?

Many targets are missing for several substances here. For example all the phenethylamines, including the 2C-derivatives target the monoamine reuptake transporters, with al 2C derivatives have only 1 line, unreadable mostly to which target the line goes, but often only to receptor targets.

Line 184. MDMA is not an addictive compound, please rewrite.

Line 188. Please support with references that 25I-NBome is extremely highly addictive, since I don’t think it is.

Line 221: to predict NPS. To predict what of NPS? See earlier comments. The model applied should be explained more clearly, and also with what aim the were used. Section 2.5. is very unclear to me

Section 2.6 is also very unclear to me. What was done, with what objective and what is presented in figure 5?

conclusions

Results and discussion are combined, which could be due to journal guidelines.

However, I am missing a clear conclusion and there is no discussion of the results. For example, how does this database related to other database in performance? How will it remain up to date? What are its limitations? For example, limited data on functional targets and no relation to dose is included, but there are more limitations that should be addressed.

Author Response

Thank you for the valuable comments concerning our manuscript, which were very useful for revising and improving our paper, as well as the important guiding significance for our researchers. We have read the comments carefully and have made relevant corrections, and we hope that the revised manuscript meets approval. The corrections in the revised manuscript are marked in red. Our point-by-point responses to the reviewer’s comments are as provided below.

Reviewer 2 Report

The manuscript details a dataset of 623 psychoactive substances from the DEA's list of controlled substances. They illustrate the consistency of their data using QSAR modeling and chemography. The data are available on the web site mentioned in the manuscript. The manuscript is straight forward, clear and of good quality. I definitely enjoyed reading it and I find the contribution useful. However, there are some points that requires the attention of the authors before publication.

Page 1 line 43-44: there is a line feed in the middle of a sentence.

Page 2 lines 49-67: “Most of the controlled compounds (…) controlled substances” describes the state of the art concerning information about psychoactive substances. There are two difficulties in this part. First, I was surprised that the authors did no mention the SMPDB (10.1093/nar/gkt1067) which is a great source of metabolic information for a large number of psychoactive drugs. It is particularly important to understand the current understanding of the mechanisms of dependence due to psychoactive substance. The second, is that at this stage I did not understood where was the focus on “new” psychoactive substances. I found plenty of well-known and not so new substances (methamphetamines for instance). Actually, I have found only the references of the DEA. Nowhere I was able to find the other datasets collected by the authors: GRAS compounds, DrugBank references, Cayman Company data, NFLIS-Drug substances and KEGG entries. I could not find or download them from the platform where I could access only to the 623 DEA referenced compounds. The authors are requested to make all the datasets available.

Page 2 lines 82-93: “Here, we present AddictedChem (…) NPS detection.” Here, one part of the message is not clear. While navigating through the structures I could find compounds presented as salts or not (lorcaserin, CAS 846589-98-8 and CAS 616202-92-7). It is never clear through the manuscript, even in the “Materials and Methods” section, how the chemical structures have been standardized. The authors are required to add a data curation paragraph in the manuscript, presumable in the “Material and Methods” section describing the processing of the chemical structures.

Page 6 to 8, table 1 and lines 235-247: “Meanwhile, the performance of (…) greatest extent”. Some of the models reported in the table 1 have extremely poor performances, likely undistinguishable from a random model on the external test set(NB::Mol2vec, LR::RDkit, RF::MHFP and ChemBERT). One of the BA is even measured below 0.5. Either, these models maybe discarded or they should be mentioned in the discussion.

Page 8 lines 252 to 254: “To show the chemical (…) distribution patterns”. I don’t understand what is a vectorization here. Which of the molecular descriptors have been used? Is it another process? The authors are requested to explain what was done and update the caption of the Figure 5 accordingly.

Page 8 Figure 5: it is not clear what is representing the picture. What is the meaning of a link? What was optimized to generate the distribution? The authors are requested to update the caption. The authors can refer to the relevant section of the “Materials and Methods”, the section 3.5.

Page 9 lines 263-265: “Overall, in the figure (…) addictive drugs”. The authors are invited to rephrase their assumption. I did not understood what is an “unknown chemical space” in this context and how the “controlled substances are likely to be developed (…) as new addictive drugs”. This is confusing.

Page 9 and 10: the authors are invited to summarize the section 3.1 in a table to ease the reading. For instance the columns could be “Source”, ”Raw dataset size”, ”Processed dataset size”, “Usage (training, test, external, etc.”, “URL”.

For the above listed reasons, I recommend to publish the manuscript after major revision.

Author Response

We are very grateful to you for taking the effort in reviewing our paper. Thank you for the patient and detailed feedback. Your valuable suggestions are of great importance in improving our work. We have carefully addressed all the comments. We have explained and modified each point accordingly. We hope you are satisfied with our responses and the new data we have provided. Changes have been made accordingly and indicated in red in the revised manuscript and in the revised supplementary information. 

Reviewer 3 Report

Interesting publication around a new platform relating to addictive compounds.

Major remark :

Difficulties in reproducing the results in the absence of precise data for the training sample and the other samples (test and external). We see statistical data based on ML methods but no chemical illustrations of success or failure for the predictions.

Author Response

We are very grateful to you for taking the effort in reviewing our paper. Your valuable suggestions are of great importance in improving our work. We have carefully addressed your comments. We have provided explanation and modified each point accordingly. We hope you are satisfied with our responses and the new data we have provided. The changes indicated in red have been made according to your suggestions. 

Round 2

Reviewer 2 Report

The authors answered all my remarks.

Author Response

Thank you very much for your suggestion.